# Methotrexate and Non-Surgical Periodontal Treatment Change the Oral–Gut Microbiota in Rheumatoid Arthritis: A Prospective Cohort Study

**DOI:** 10.3390/microorganisms12010068

**Published:** 2023-12-29

**Authors:** Sicília Rezende Oliveira, José Alcides Almeida de Arruda, Jôice Dias Corrêa, Valessa Florindo Carvalho, Julliane Dutra Medeiros, Ayda Henriques Schneider, Caio Cavalcante Machado, Letícia Fernanda Duffles, Gabriel da Rocha Fernandes, Débora Cerqueira Calderaro, Mario Taba Júnior, Lucas Guimarães Abreu, Sandra Yasuyo Fukada, Renê Donizeti Ribeiro Oliveira, Paulo Louzada-Júnior, Fernando Queiroz Cunha, Tarcília Aparecida Silva

**Affiliations:** 1Department of Oral Surgery, Pathology and Clinical Dentistry, School of Dentistry, Federal University of Minas Gerais, Belo Horizonte 31270-901, MG, Brazil; siciliarezende@gmail.com (S.R.O.); alcides_almeida@hotmail.com (J.A.A.d.A.); 2Department of Dentistry, Pontifical Catholic University, Belo Horizonte 30535-901, MG, Brazil; jo.dias.c@gmail.com; 3Department of Oral and Maxillofacial Surgery and Periodontology, School of Dentistry of Ribeirão Preto, University of São Paulo, Ribeirão Preto 14040-900, SP, Brazil; valessa@outlook.com (V.F.C.); mtaba@usp.br (M.T.J.); 4Department of Biology, Federal University of Juiz de Fora, Juiz de Fora 36036-900, MG, Brazil; jdutramedeiros@gmail.com; 5Department of Pharmacology, Ribeirão Preto Medical School, University of São Paulo, Ribeirão Preto 14040-900, SP, Brazil; ayda.hs@usp.br (A.H.S.); fdqcunha@fmrp.usp.br (F.Q.C.); 6Division of Clinical Immunology, Ribeirão Preto Medical School, University of São Paulo, Ribeirão Preto 14040-900, SP, Brazil; caiocavalcant@gmail.com (C.C.M.); reneimuno@yahoo.com.br (R.D.R.O.); plouzada@fmrp.usp.br (P.L.-J.); 7Department of BioMolecular Sciences, School of Pharmaceutical Science, University of São Paulo, Ribeirão Preto 14040-900, SP, Brazil; leticiaduffles@gmail.com (L.F.D.); sfukada@usp.br (S.Y.F.); 8René Rachou Research Center, Oswaldo Cruz Fundation, Belo Horizonte 30190-002, MG, Brazil; fernandes.gabriel@gmail.com; 9Department of Locomotor Apparatus, Faculty of Medicine, Federal University of Minas Gerais, Belo Horizonte 31270-901, MG, Brazil; dccalderaro@gmail.com; 10Department of Child and Adolescent Oral Health, School of Dentistry, Federal University of Minas Gerais, Belo Horizonte 31270-901, MG, Brazil; lucasgabreu01@gmail.com

**Keywords:** gut microbiome, methotrexate, microbiota, mouth, periodontitis, rheumatoid arthritis

## Abstract

This study evaluated the changes in the composition of oral–gut microbiota in patients with rheumatoid arthritis (RA) caused by methotrexate (MTX) and non-surgical periodontal treatment (NSPT). Assessments were performed at baseline (T0), 6 months after MTX treatment (T1), and 45 days after NSPT (T2). The composition of the oral and gut microbiota was assessed by amplifying the V4 region of the 16S gene from subgingival plaques and stools. The results of the analysis of continuous variables were presented descriptively and non-parametric tests and Spearman’s correlation were adopted. A total of 37 patients (27 with periodontitis) were evaluated at T0; 32 patients (24 with periodontitis) at T1; and 28 patients (17 with periodontitis) at T2. MTX tended to reduce the alpha diversity of the oral–gut microbiota, while NSPT appeared to increase the number of different species of oral microbiota. MTX and NSPT influenced beta diversity in the oral microbiota. The relative abundance of oral microbiota was directly influenced by periodontal status. MTX did not affect the periodontal condition but modified the correlations that varied from weak to moderate (*p* < 0.05) between clinical parameters and the microbiota. MTX and NSPT directly affected the composition and richness of the oral–gut microbiota. However, MTX did not influence periodontal parameters.

## 1. Introduction

Rheumatoid arthritis (RA) is an autoimmune disease characterized by synovial inflammation and bone erosion, resulting in joint damage [1]. The etiology of the disease involves genetic and environmental factors, as well as immune system dysfunction [2]. Interactions between microorganisms and the immune system are also implicated in the progression of RA [3]. Individuals with RA have an increased incidence of periodontitis [4]. Periodontitis is a multifaceted and dynamic inflammatory condition influenced by multiple factors, among them, microbial dysbiosis, host response, environmental influences, tissue remodeling, systemic connections, and patient-centered impacts, resulting in the disruption of the supportive structures surrounding the teeth [1,5]. Previous studies have proposed a two-way model where RA exacerbates periodontal destruction and the resulting inflammatory response worsens RA progression [6,7].

The recent literature has reported dysbiosis of oral [8,9,10,11] and gut [8,12,13] microbiota in individuals with RA. In turn, the increase in alpha diversity is more common in the oral cavity, while a decrease or stability of diversity is observed in the gut microbiota [13]. These findings support the theory that RA represents a chronic inflammatory state that can be triggered or exacerbated by the overgrowth of pathogenic bacteria which are often able to modulate the immune response [3,8,13].

Disturbances in the oral and gut microbiomes in RA have also been associated with a loss of tolerance against autoantigens and an increase in inflammatory events that promote joint damage [14]. Among the numerous bacterial pathogens potentially involved in RA, the Gram-negative bacterium *Porphyromonas gingivalis* plays an important role due to its ability to induce citrullination, which elicits or modifies the host immune response [15]. Other periodontal pathogens, including *Aggregatibacter actinomycetemcomitans* and *Prevotella intermedia*, play a role in the pathogenesis of RA by inducing hypercitrullinization [16,17] and can be identified both in subgingival plaque and in the serum of individuals with RA [18]. Within the context of the gut microbiota, *Prevotella copri* is also immune-relevant in RA pathogenesis [19].

The folic acid antagonist methotrexate (MTX) is currently the anchor drug for RA treatment—either as a single agent or in combination with other disease-modifying antirheumatic drugs (DMARDs) [20]. Oral MTX can be partially metabolized by the gut microbiota; in turn, the microbiome may be useful in predicting the response to MTX and may also be a potential target in the attempt to improve the response to this drug [21]. Studies in mice have revealed that MTX affects the abundance of some species in the gut and protects against the periodontal bone loss associated with arthritis and periodontal infection [22,23]. However, the combined effects of MTX and of short- and long-term non-surgical periodontal treatment (NSPT) on the oral–gut microbiota in individuals with RA, along with the association with clinical parameters, are unknown. The scarcity of literature highlights the need for more comprehensive studies investigating the simultaneous effects of MTX on the oral–gut microbiota in individuals with RA. Specific changes in microbial diversity, composition, and function in response to MTX treatment and how these changes may correlate with clinical outcomes and disease progression have barely been explored [8,13].

The purpose of the present longitudinal study was to investigate the changes in the oral–gut microbiota composition of individuals with RA caused by MTX and NSPT and their relationship with clinical and biochemical parameters. Our hypothesis was that MTX and NSPT would change the diversity of the oral–gut microbiota.

## 2. Materials and Methods

### 2.1. Study Design, Setting, and Ethical Clearance

This prospective cohort study was performed at the Rheumatology Outpatient Clinic of the Ribeirão Preto Medical School Hospital, São Paulo, Brazil, between August 2018 and December 2019. The guidelines for Strengthening the Reporting of Observational Studies in Epidemiology (STROBE) were followed [24] and the study was approved by the Institutional Ethics Committee (#05934818.3.0000.5440). Patients gave written informed consent to participate in the study and their anonymity was preserved in accordance with the Declaration of Helsinki.

### 2.2. Participants and RA Diagnostic Criteria

Individuals aged ≥18 years and who had at least eight teeth, diagnosed with RA based on 2010 ACR/EULAR classification criteria [25], were included in the study. The patient should have experienced arthralgia for at least six weeks and clinically detected synovitis in at least one joint to be classified as having RA. The diagnosis is made when the patient achieves a total score of ≥6 (out of a possible 10) based on the number of painful and swollen joints, inflammatory markers (e.g., C-reactive protein (CRP) and erythrocyte sedimentation rate (ESR)), as well as the presence of autoantibodies (e.g., rheumatoid factor and anti-citrullinated protein antibodies (ACPA)) in the absence of an alternative diagnosis that would better explain the synovitis [25]. Individuals with symptom duration ≤24 months were classified as “early RA” and those with >24 months of symptoms were classified as “established RA” [26].

The exclusion criteria were: individuals diagnosed with another rheumatic disease (e.g., Sjögren’s syndrome, systemic lupus erythematosus, or primary osteoarthritis), individuals who received DMARDs other than MTX as the initial treatment or who were already undergoing treatment with MTX or any other DMARDs before the first assessment, and individuals with diabetes mellitus or malignant neoplasms. Individuals undergoing treatment for periodontal disease within the last 6 months, individuals wearing orthodontic appliances, those who took antibiotics within the last 3 months, and pregnant or lactating women were also excluded [27].

### 2.3. Data Assessment and Collection

The patients were evaluated at three time points. The first assessment (T0) was performed before starting MTX treatment and the second (T1) was performed six months after starting MTX treatment; at T1, after the second evaluation, the patients underwent NSPT. The third evaluation (T2) was performed 45 days after NSPT (Figure 1).

### 2.4. Clinical and Laboratory RA Assessment

The following data were collected from patients’ records: rheumatoid factor, ESR, CRP, ACPA, pain measured with a visual analog scale (VAS), time of symptoms, and time of treatment. RA disease activity was determined by calculating the ESR-DAS 28 score (disease activity score 28) and was assessed at T0 and T1. At T1, the responsiveness of individuals to MTX treatment was also evaluated. Individuals who regularly used MTX orally for at least three months at a dose ≥15 mg/week and who exhibited a decrease in DAS 28 ≥ 1.2, and a final DAS 28 ≤ 3.2 were considered to be responsive to treatment. Individuals who did not meet these criteria were considered to be unresponsive [28].

### 2.5. Periodontal Evaluation and Treatment

Periodontal status was evaluated by two calibrated examiners (V.F.C. and T.A.S.), who used a periodontal probe [model PCP15, Hu-Friedy^®^; Chicago, IL, USA] at three time points (T0, T1, and T2). Weighted kappa agreement tests for periodontal clinical parameters revealed intra- and inter-examiner values ≥ 92%. The following parameters were recorded: plaque index, probing depth, clinical attachment loss (CAL), and bleeding on probing (BOP). Participants were classified as individuals without periodontitis or with mild, moderate, and severe periodontitis [29]. In T1, a single session of NSPT (full-mouth scaling and root planing) was performed with ultrasound and manual Gracey curettes in all individuals. Oral hygiene instructions were also provided.

### 2.6. Subgingival Plaque and Stool Samples for Microbiome Analysis

Subgingival plaque samples were collected with sterile absorbent paper points [Dentsply Sirona; New York, NY, USA] inserted for 1 min at five sites with the deepest probing depth. Next, all material was pooled, stored in a sterile tube containing 500 µL of sterile distilled water, and centrifuged at 3000× *g* for 5 min. The paper points were discarded, and the pellet was kept at −80 °C until DNA extraction [9]. Fresh stool samples were collected from patients using an OMNI gene^®^ GUT|OM-200 kit [DNA Genotek Inc.; Ottawa, ON, Canada] for gut microbiome profiling according to the manufacturer’s instructions. Samples were stored at 8 °C until DNA extraction.

#### 2.6.1. DNA Extraction and 16S rRNA Amplicon Library Preparation and Sequencing

DNA was extracted from subgingival and stool samples using the DNeasy PowerSoil Pro kit [QIAGEN; Hilden, Germany]. DNA concentration was evaluated spectrophotometrically using a NanoDrop™ spectrophotometer [Thermo Fisher; Waltham, MA, USA]. The resulting purified DNA was stored at −20 °C until use.

For the subgingival samples, the V4 hypervariable region of the 16S rDNA gene was amplified using the 341F and 806R primer pair [30,31]. The primers were modified to obtain an overhang sequence complementary to the Nextera XT DNA Library Prep Kit [Illumina; San Diego, CA, USA] and were applied to a final volume of 25 μL containing 1× NEBNext^®^ Q5 Hot Start HiFi PCR Master Mix [New England Biolabs; Ipswich, MA, USA], a 0.2 μM primer, and 12.5 ng of DNA. Cycling conditions were 95 °C for 3 min followed by 30 cycles of 95 °C (30 s), 55 °C (30 s), and 72 °C (30 s), and a final extension at 72 °C (5 min).

For the stool samples, the V4 region of the 16S rDNA gene was amplified with the region-specific 515F primers [32] and 806R [33] constructed with barcodes according to the method proposed by the Earth Microbiome Project (https://earthmicrobiome.org/protocols-and-standards/16s/; accessed on 1 February 2021). PCR was performed in a final 25 μL volume containing 1× PCR buffer, 0.2 mM dNTP, 2 mM MgSO_4_, 0.2 μM of primers, 1U Platinum™ Taq DNA Polymerase High Fidelity [Thermo Fisher; Waltham, MA, USA], and 20–30 ng of DNA. Cycling conditions were 94 °C for 3 min followed by 30 cycles at 94 °C (45 s), 57 °C (60 s), and 68 °C (45 s), and a final extension at 68 °C (10 min).

All libraries were purified using AMPure XP beads [Beckman Coulter; Brea, CA, USA] according to the Illumina 16S metagenomic sequencing library protocol (https://www.illumina.com/content/dam/illuminasupport/documents/documentation/chemistry_documentation/16s/16s-metagenomic-library-prep-guide-15044223-b.pdf; accessed on 1 February 2021). The size of the libraries was checked by the Bioanalyzer DNA 1000 Assay [Agilent; Santa Clara, CA, USA]. Quantifications were performed with the KAPA Library Quantification Kit Illumina [Roche; Basel, Switzerland], and further DNA libraries were normalized and pooled. The libraries were paired-end sequenced (2 × 300) in a single run on the Miseq platform [Illumina; San Diego, CA, USA]. All sequencing was performed on the Next Generation Sequencing platform of the René Rachou Institute (CPqRR/FIOCRUZ, Belo Horizonte, Brazil).

#### 2.6.2. Read Processing and Taxonomic Classification

The raw read files were processed in the R environment using the DADA2 package [34]. The forward and reverse sequences were trimmed to 180 and 130 bases, respectively. Readings containing more than two expected errors were removed. The errors of the filtered sequences were corrected by the algorithm and were joined to form the Amplicon Sequence Variants. The chimeric sequences were also removed and a sample count table was generated. Taxonomic classification was performed using the TAG.ME package and 515F-806R model [35].

#### 2.6.3. Data Analysis

The environmental beta diversity was measured using the principal coordinate analysis function of the “ade4 R” package [36] based on the Jensen–Shannon divergence calculated by the philentropy R package [37]. PERMANOVA was performed using 999 permutations to test the impact of categorical variables on beta diversity, and alpha diversity was measured by the Shannon and Simpson indices. The PERMANOVA test and alpha diversity calculations were performed using the vegan R package. Differential abundance between groups was tested with the DESeq2 R package [38] using the absolute read counts for each taxonomic entry in a Wald test. Differences supported by adjusted values of *p* < 0.01 were selected for discussion.

### 2.7. Statistical Analysis

The Statistical Package for the Social Sciences (SPSS) (version 25.0, Armonk, New York, NY, USA) and the GraphPad Prism (version 7.00, La Jolla, CA, USA) were employed for the statistical analyses. Descriptive analysis was applied to clinicodemographic aspects and periodontal data. The Shapiro–Wilk test revealed that continuous data exhibited a non-normal distribution (*p* < 0.001). Bivariate analysis involving continuous data was performed with the Wilcoxon test. The results of the analysis were presented as median, minimum, and maximum. The chi-square test was adopted for bivariate analysis involving categorical data. Spearman’s correlation coefficient was used to measure the correlation between bacterial content and quantitative variables. For all analyses, the level of significance was set at <0.05.

## 3. Results

### 3.1. Clinicodemographic Data

A total of 37 individuals diagnosed with RA were included in this study; of these, 22 were in the early stage of the disease, and 15 were in the established stage. The median age was 51 years, with most individuals being females (89.2%) who had never smoked (59.5%) or consumed alcohol (83.8%). Regarding RA disease activity, the median DAS 28 observed was 4.99 and many individuals had high disease activity. The median treatment time was three months. Among the individuals included, most had a diagnosis of periodontitis. The clinicodemographic characteristics of the included individuals are presented in Table 1.

### 3.2. NSPT Modified Periodontal Parameters

At T0, 37 individuals were evaluated. After 6 months of treatment with MTX alone (T1), 32 individuals were reevaluated. There were no significant reductions in isolated periodontal parameters and no changes in periodontitis severity (*p* > 0.05), except for BOP, for which a significant increase was observed at T1 (*p* < 0.05). Regarding disease activity, a reduction in DAS 28 was observed after treatment with MTX (T1), albeit not significant (*p* > 0.05). At T1, 24 individuals with periodontitis and eight without periodontitis received NSPT. Forty-five days after NSPT (T2), 28 individuals were reevaluated; of these, 17 exhibited periodontitis and 11 did not. There was a significant reduction in periodontal parameters, such as probing depth, CAL, BOP, and plaque index, in addition to a significant reduction in the number of individuals with moderate-stage periodontitis (*p* < 0.05) (Table 2). Individuals who had missed the follow-up appointment (*n* = 5), who had undergone NSPT periodontal treatment at another service (*n* = 3), and those who had used a systemic antibiotic during the study (*n* = 1) were excluded.

### 3.3. MTX and NSPT Affected Subgingival Microbial Richness and Diversity

Comparisons of subgingival microbial diversity were performed for individuals with RA at recruitment and at two follow-up times (T1 and T2). A trend towards decreased alpha diversity was observed after MTX treatment. In contrast, successful NSPT appeared to also influence alpha diversity, as shown by an increase in the number of different species of oral microbiota (Figure 2A). In comparisons related to beta diversity (i.e., alteration in the structure of the oral microbiota), MTX and NSPT seemed to have an influence on the bacterial composition of individuals. However, no clustering of microbiota samples was observed among the three different assessments (Figure 2B).

### 3.4. Periodontitis Influenced the Relative Abundance of Different Genera in the Oral Microbiota

The composition of the oral microbiota differed significantly between groups of individuals according to their periodontal status. Among individuals with severe periodontitis, there was an increase in the relative abundance of the genera *Anaeroglobus*, *Eubacterium.saphenum*, *Defluviitaleaceae.UCG.011*, *Eubacterium.yurii*, *Pseudoramibacter*, *unclassified.Bacteroidaceae*, *Selenomonas*, *Desulfobulbus*, *Eubacterium.nodatum*, and *Mycoplasma*. Also, *Dialister* bacteria were increased in both moderate and severe periodontitis. Among individuals with moderate periodontitis, a genus of *unclassified Archaea*, *Eikenella*, and *Staphylococcus* showed increased relative abundance. In contrast, among individuals without periodontitis, only an increase in the *unclassified Weeksellaceae* was observed (Figure 3).

### 3.5. MTX Treatment and Gut Microbiome Diversity

Similar to what was observed in the oral microbiota, MTX treatment decreased alpha diversity in the gut microbiota (Figure 4A). Regarding beta diversity, MTX treatment punctually altered the structure of the gut microbiota (Figure 4B). Individuals who did not respond to MTX treatment had a significantly higher relative abundance of *Abiotrophia* compared to those who responded to treatment (Appendix A). No additional differential abundance within taxonomic groups of the gut microbiota was associated with MTX treatment.

### 3.6. Periodontal Parameters and RA Activity Influenced the Oral and Gut Microbiota

At the baseline evaluation of the participants, correlations of the oral and gut microbiome with the periodontal clinical parameters were observed (Figure 5). The plaque index was directly correlated with *Prevotella.2* (*rho* = 0.445; *p* = 0.006) and *Campylobacter* (*rho* = 0.527; *p* < 0.001), and inversely correlated with *Alloprevotella* (*rho* = −0.363; *p* = 0.029). BOP was directly correlated with *Fusobacterium* (*rho* = 0.408; *p* = 0.013). On the other hand, the number of teeth was inversely correlated with *Aggregatibacter* (*rho* = −0.456; *p* = 0.005) and *Prevotella.2* (*rho* = −0.383; *p* = 0.020). There was a direct correlation between DAS 28 and *Campylobacter* (*rho* = 0.390; *p* = 0.018). Also, treatment time showed an inverse correlation with *Leptotrichia* (*rho* = −0.388; *p* = 0.030) and *Veillonella* (*rho* = −0.370; *p* = 0.040). A direct correlation between CRP and *Leptotrichia* was also observed (*rho* = 0.379; *p* = 0.032).

Regarding the gut microbiome, there was an inverse correlation between BOP and the *unclassified Clostridiales* (*rho* = −0.373; *p* = 0.032), *Ruminococcaceae.UCG.002* (*rho* = −0.416; *p* = 0.016), and *Ruminococcaceae.UCG.005* (*rho* = −0.359; *p* = 0.039). Probing depth correlated directly with *Bacteroides* (*rho* = 0.410; *p* = 0.014) and inversely with *Ruminococcaceae.UCG.005* (*rho* = −0.365; *p* = 0.031), *Desulfovibrio* (*rho* = −0.342; *p* = 0.043), and the *Ruminococcaceae.NK4A214.group* (*rho* = −0.348; *p* = 0.040).

CAL and bacteria of the genera *Bacteroides* (*rho* = 0.375; *p* = 0.034) and *Agathobacter* (*rho* = 0.367; *p* = 0.038) were also directly correlated. Direct correlations of rheumatoid factor (*rho* = 0.422; *p* = 0.028) and time of symptoms (*rho* = 0.425; *p* = 0.019) with the *Lachnospiraceae.NK4A136.group* were observed. Time of symptoms exhibited a direct correlation with the *Ruminococcaceae.UCG.003* (*rho* = 0.416; *p* = 0.021) and with the *Eubacterium.eligens.group* (*rho* = 0.377; *p* = 0.039). In contrast, an inverse correlation was observed between the time of treatment and *Tyzzerella* (*rho* = −0.384; *p* = 0.032). Direct correlations of CRP with *Butyricimonas* (*rho* = 0.385; *p* = 0.035), *Eubacterium.eligens* (*rho* = 0.476; *p* = 0.007)*,* and *unclassified Lachnospiraceae* (*rho* = 0.466; *p* = 0.009) were observed. The magnitude of all observed correlations varied from weak to moderate and were statistically significant (*p* < 0.05).

### 3.7. MTX Treatment Defined New Associations of Periodontal and RA Parameters with Oral and Gut Bacteria

Six months after MTX treatment, a change in correlations between clinical parameters and oral and gut bacteria was observed, with the emergence of new correlations (Figure 6). BOP was directly correlated with oral bacteria of the genus *Streptococcus* (*rho* = 0.361; *p* = 0.042) and *Capnocytophaga* (*rho* = 0.373; *p* = 0.035). Plaque index was also directly correlated with *Streptococcus* (*rho* = 0.355; *p* = 0.046) and inversely correlated with *Porphyromonas* (*rho* = −0.418; *p* = 0.017) and *Fusobacterium* (*rho* = −0.413; *p* = 0.018). The number of teeth was inversely correlated with *Capnocytophaga* (*rho* = −0.396; *p* = 0.024).

Regarding the gut microbiome, the number of teeth showed an inverse correlation with the bacteria of the genus *unclassified Lachnospiraceae* (*rho* = −0.553; *p* = 0.003), *Faecalibacterium* (*rho* = −0.398; *p* = 0.043), *Sutterella* (*rho* = −0.451; *p* = 0.0207), *Odoribacter* (*rho* = −0.419; *p* = 0.032), *Oscillibacter* (*rho* = −0.501; *p* = 0.008), *Ruminococcaceae.UCG.002* (*rho* = −0.424; *p* = 0.030), *unclassified Ruminococcaceae* (*rho* = −0.462; *p* = 0.017), *Ruminococcus.torques.group* (*rho* = −0.445; *p* = 0.022), *Eubacterium.coprostanoligenes.group* (*rho* = −0.463; *p* = 0.017), *Butyricicoccus* (*rho* = −0.432; *p* = 0.027), *Lachnoclostridium* (*rho* = −0.584; *p* = 0.001), *Ruminiclostridium.9* (*rho* = −0.439; *p* = 0.024), *Lachnospiraceae.NK4A136.group* (*rho* = −0.485; *p* = 0.011), and *Subdoligranulum* (*rho* = −0.503; *p* = 0.008). In contrast, a direct correlation was observed between the number of teeth and *Prevotella.9* (*rho* = 0.448; *p* = 0.021). An inverse correlation between BOP and *Ruminococcaceae.UCG.003* was also observed (*rho* = −0.395; *p* = 0.041). DAS 28 showed a direct correlation with *Tyzzerella* (*rho* = 0.444; *p* = 0.022), *Coprococcus.3* (*rho* = 0.489; *p* = 0.011), *Ruminococcaceae.UCG.005* (*rho* = 0.528; *p* = 0.005), and *Roseburia* (*rho* = 0.404; *p* = 0.040). Importantly, after treatment with MTX, new correlation tests were performed. A priori, inverse correlations between the number of teeth and oral bacteria were observed. No correlation between number of teeth and gut bacteria was observed. After the MTX regimen, numerous new inverse correlations were observed between the number of teeth and gut bacteria. Likewise, initially, DAS 28 only showed a direct correlation with an oral bacterium (*Campylobacter*). After treatment with MTX, numerous direct correlations were observed between DAS 28 and gut bacteria (e.g., *Tyzzerella, Roseburia*). The magnitude of all observed correlations varied from weak to moderate and were statistically significant (*p* < 0.05).

## 4. Discussion

This study demonstrated that a 6-month MTX treatment reduced alpha diversity and modified the structure of the oral and gut microbiota in RA patients. NSPT influenced the diversity of the oral microbiota. Moreover, NSPT improved periodontal parameters individually, whereas MTX treatment had no impact on the periodontium. Finally, MTX modified the relationship between periodontal and RA clinical factors.

Without early and efficient treatment, RA can result in progressive disability and extra-articular symptoms, leading to increased morbidity and mortality [2]. MTX targets the inflammation pathways responsible for joint swelling and damage [20]. However, a significant non-response rate of 30 to 40%, as well as the impossibility of predicting the patient’s response to treatment, are evident impasses [39]. The recent literature has indicated multiple factors involved with MTX responsiveness, including the participation of the gut microbiota in drug metabolism and efficiency [40,41]. Understanding how the microbiota affects the response to MTX and, in turn, what impact this drug has on the microbiota is important not only because the modulation of the gut microbiota may offer a novel therapeutic or preventive approach for patients with RA, but also because it may be helpful in predicting the response to treatment. Herein, we observed that a decrease in the alpha diversity of both the oral and gut microbiota occurred after MTX treatment. Zhang et al. [8] reported that dysbiosis was detected in the gut and oral microbiomes of RA patients and was partially modified after RA treatment. Hence, these findings have not been uniform [10,15], perhaps due to the different study designs, sampling sites, or assays used. Another report also noted that the microbiota of RA patients was sensitive to MTX, to changes in gut bacterial taxa, and to gene family abundance [42]. The fine line that separates this effect from dysbiosis may depend on the intrinsic ability to metabolize the drug or on factors such as the pharmacological combination, the dose administered, or the length of time the drug is prescribed [40].

At baseline, numerous positive and negative correlations were observed between oral and gut bacteria, as well as correlations with clinical parameters of both periodontitis and RA. It is interesting to note the direct correlation between the clinical parameters, namely plaque index and DAS 28, and the pathobiont oral bacterium *Campylobacter*, which has been previously reported to be abundant in individuals with RA and periodontitis [43]. In the present study, an inverse correlation was observed between the oral pathogenic bacteria *Leptotrichia* and treatment time, as well as a positive correlation with CRP. In the context of systemic lupus erythematosus, we demonstrated direct correlations between *Leptotrichia* and levels of pro-inflammatory interleukins [44]. Sher et al. [45] demonstrated that *Leptotrichia* species were predominant in individuals with recent-onset RA, thus indicating a consistent role of this oral bacterium in the onset of RA [14]. Several other correlations have also been demonstrated between gut bacteria and periodontal parameters elsewhere [10,46]. For example, gut bacteria of the genus *Bacteroides*, which have only been observed abundantly in individuals with RA and which may be associated with disease progression [46], were directly correlated with clinical parameters, probing depth, and CAL. Conversely, an inverse correlation was noted between *Bacteroides* and the oral bacterium *Capnocytophaga*, which a priori was associated with lower probing depth in individuals with RA [10].

In the present study, after treatment with MTX, changes occurred in the correlation network, mainly in gut bacteria, which exhibited numerous new direct correlations with DAS 28 and inverse correlations with the number of teeth. It is important to note that correlations between pathogenic bacteria such as *Tannerella* and *Treponema* have not been previously observed after MTX treatment. On the contrary, new correlations with the health-related bacteria *Streptococcus*, *Capnocytophaga*, and *Tyzzerella* emerged after MTX treatment. This scenario of new correlations reinforces the evidence that changes in microbiota diversity and disease parameters may occur after MTX treatment. In fact, the mechanism of response to MTX in patients with RA is likely to be associated with the catabolic capacity of the drug in the gut microbiota [41]. Although no microbiological signature regarding response to MTX was observed in our study, individual biological variations, disease parameters, and genetic susceptibility may explain the responsiveness to MTX treatment.

Relevant data refer to the impact of MTX treatment on periodontal parameters [22]. Contrary to the results observed in an animal model of arthritis in which MTX treatment protected against alveolar bone loss [23], in the present study there was no apparent effect of MTX on the periodontium. Other clinical studies have shown similar results, i.e., stability of periodontal parameters in individuals with moderate to severe periodontitis after 16 months of antirheumatic treatment [47]. In contrast, an improvement in periodontal inflammation parameters following treatment with conventional DMARDs has been reported elsewhere [48,49]. Evidence indicates that DMARDs, including MTX, are able to improve the periodontal condition of individuals with RA and periodontitis; hence, to observe such an effect on a short-term basis is unfeasible [50,51]. The short follow-up period, limited sample size, and initial 6-month period without combined professional oral prophylaxis as part of our study design certainly explain the maintenance of individual periodontal parameters before and after MTX treatment. Thus, studies with more robust samples and longer follow-up times are necessary to confirm the data.

NSPT resulted in the improvement of periodontal parameters and also influenced the diversity of our patients’ oral microbiota. Another report has also observed the effect of NSPT among individuals with RA [46]. Studies investigating the influence of NSPT on the subgingival microbiota of systemically healthy individuals with chronic periodontitis initially observed a decrease in alpha diversity after two and six weeks of NSPT, which was completely restored after 12 weeks [52]. In our study, a decrease, albeit non-significant, in microbial diversity was also demonstrated when compared to baseline. However, a change in this microbiota was noticed after NSPT when compared to the time after the antirheumatic treatment. These findings corroborate the notion that greater microbiota diversity is associated with increased ecosystem resilience and a healthier condition [53,54].

It was also observed that NSPT individually modified the structure of the oral microbiota; however, it was not possible to identify a pattern in these changes due to the non-formation of clusters. Changes in the composition of the microbiota after NSPT have also been reported in the literature. This reflects a significant decrease in the relative abundance of periodontopathogenic bacteria such as *Porphyromonas* and *Treponema* species, and an increase in health-associated bacteria, such as *Streptococcus* and *Rothia* species, shortly after NSPT, but these changes were gradually reversed within 12 weeks after treatment [52]. Other studies that analyzed the salivary microbiota after NSPT in systemically healthy individuals with aggressive periodontitis documented a trend towards reduced diversity measured by the Shannon index after three and six months of NSPT compared to baseline [55]. It has also been suggested that other adjunctive periodontal therapies such as antimicrobial photodynamic treatment may lead to a reduction in periodontopathogenic bacteria in individuals with RA [56]. Nonetheless, it is important to highlight the individuality of each patient’s oral microbiota and that there is probably no single composition that represents a healthy periodontal state. Additionally, recovery from periodontal disease appears to reflect a shift from a personalized disease state to a personalized healthy state. While there is consensus that specific communities must change with the response to disease, there may not be a “healthy amount” of these bacteria that is consistent across individuals [57].

The severity of periodontitis of the patients evaluated here was associated with a greater relative abundance of some pathogenic bacteria. In this line, changes in the subgingival microbial profile of individuals with RA directly associated with the severity of periodontitis were demonstrated [11]. Conversely, other authors have not demonstrated significant differences between the subgingival microbial profile of individuals with RA and the classification of periodontitis. Nevertheless, individual periodontal parameters such as deep periodontal pockets have been directly associated with a greater abundance of Gram-negative anaerobic pathogens such as *Selenomonas* [10], which was also observed more abundantly in our study. Likewise, bacteria of the genus *Anaeroglobus* have been associated with severe periodontitis. Bacteria of this genus have been correlated with an increase in the number of swollen and painful joints and with the levels of circulating rheumatoid factor and ACPA in individuals with RA [9,14,45]. These findings indicate a possible role of this oral bacterium, i.e., its increase in severe periodontitis, in the progression of RA [9,14,45]. We also showed that individuals with severe periodontitis had a higher abundance of the *Desulfobulbus* genus when compared to those without periodontitis. This aligns with former studies reporting that the *Desulfobulbus* genus was increased in individuals with RA and more severe periodontitis [11]. Other authors have also identified the severity of periodontitis as a determining factor in defining the diversity of the subgingival microbiota [45] and have identified pathogenic bacteria of the “red complex” associated with more severe periodontitis in individuals with RA [58]. These findings suggest that differences in the relative abundance of the subgingival microbiota characterize more severe forms of periodontitis and do not represent a specific signature for the oral microbiota of RA [45].

The current study has shortcomings that should be acknowledged. The first concerns the absence of a control group of individuals who had not been exposed to MTX and NSPT for comparison, precluding the conclusion that changes in oral–gut microbiota were due solely to the exposures being studied, rather than to other factors. The relatively small sample size and the limitations of periodontitis classifications in terms of their complex nature for implementation in clinical practice should also be highlighted. Several aspects such as lifestyle (e.g., physical activity and stress management), dietary patterns (e.g., cultural and regional differences), and individual variability (e.g., genetic factors) can have a profound impact on shaping microbial communities in the oral–gut axis and cannot be ruled out [19]. Taken together, these factors underscore the need for cautious interpretation, particularly with regard to the generalizability of the results. Nonetheless, this study offers a valuable contribution to the understanding of the complex interplay between MTX treatment, periodontal health, and the oral–gut microbiota in individuals with RA. Of clinical relevance, the importance of evaluating changes at multiple time points to capture the dynamic nature of the microbiota under the influence of periodontal treatments supports the idea of performing NSPT at the beginning of RA treatment with MTX.

## 5. Conclusions

Based on the findings of the present study, MTX treatment reduced the diversity and richness of the oral and gut microbiota and also modified the correlations between the bacterial and clinical parameters of RA and periodontitis. Periodontal treatment resulted in changes in the oral microbiota and the severity of periodontitis influenced the relative abundance of oral bacteria. However, antirheumatic treatment had no effect on periodontal clinical parameters. Defining the mechanisms by which MTX alters the oral and gut microbiota may be useful for predicting and monitoring the response to treatment. Future studies on robust samples with a suitable experimental design are necessary to detect subtle effects and validate our data.

## Figures and Tables

**Figure 1 microorganisms-12-00068-f001:**
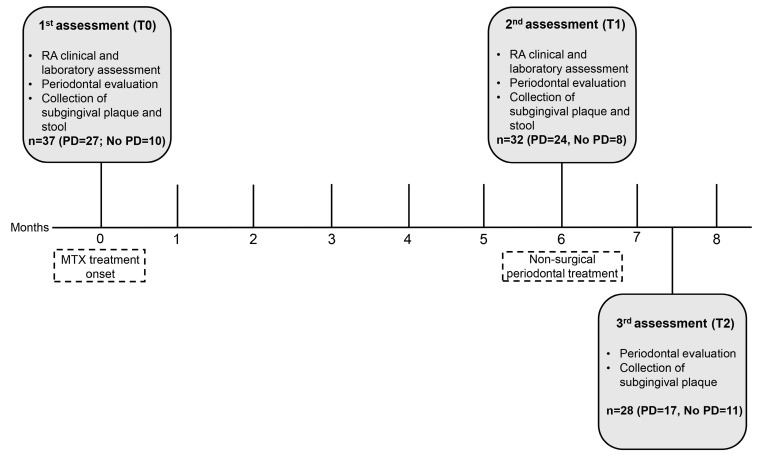
Diagram representing data collection. MTX: methotrexate; PD: periodontitis.

**Figure 2 microorganisms-12-00068-f002:**
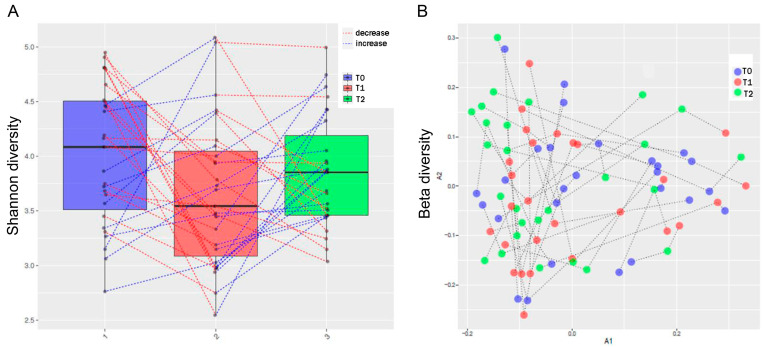
Diversity of oral microbiota. (**A**) Shannon alpha diversity index. (**B**) Beta diversity index. The blue dashed lines connect the samples that showed increased diversity in relation to the immediately previous assessment (T1 ˃ T0; T2 ˃ T1) and the red dashed lines demonstrate the decrease in diversity in relation to the immediately previous assessment (T1 < T0; T2 < T1).

**Figure 3 microorganisms-12-00068-f003:**
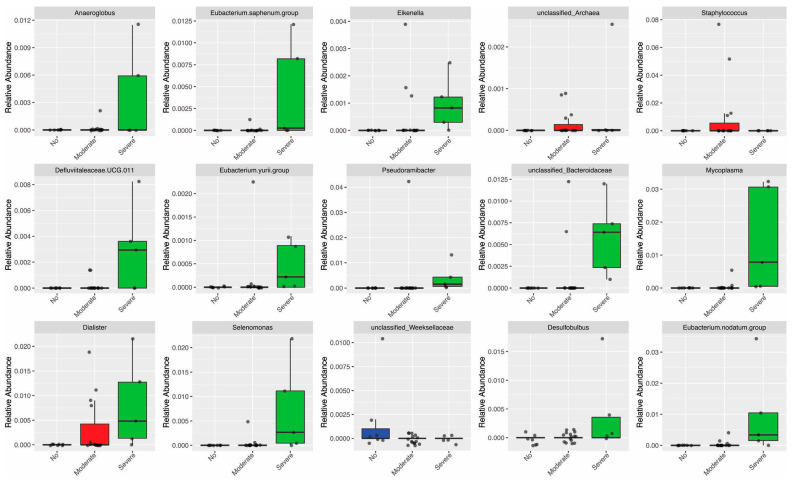
Genus abundance that differed between periodontal status groups (DESeq2).

**Figure 4 microorganisms-12-00068-f004:**
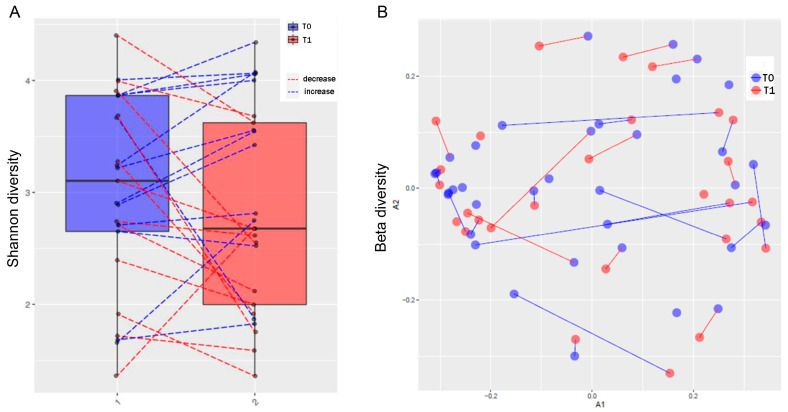
Diversity of gut microbiota. (**A**) Shannon alpha diversity index. (**B**) Beta diversity index. The blue dashed lines connect the samples with increased diversity compared to the immediately previous assessment (T1 ˃ T0; T2 ˃ T1), and the red dashed lines demonstrate the decrease in diversity compared to the immediately previous assessment (T1 < T0; T2 < T1).

**Figure 5 microorganisms-12-00068-f005:**
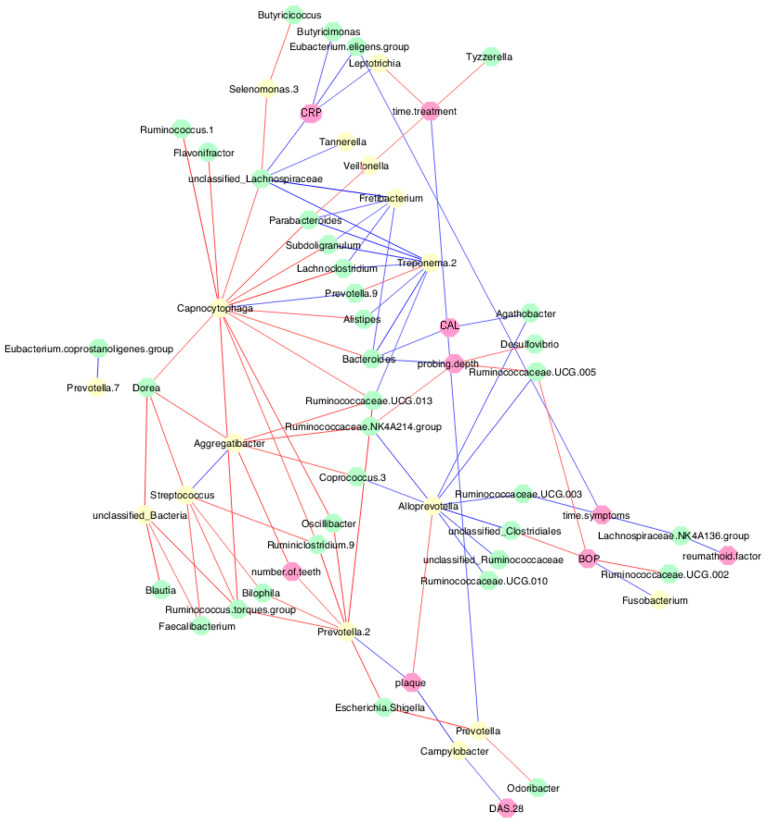
Correlations between oral and gut microbiota and clinical data before methotrexate (MTX) treatment. Spearman’s correlation coefficient was used to measure the correlation between bacterial content and quantitative variables. The blue lines represent direct correlations, and the red lines represent inverse correlations. Yellow circles represent oral bacteria, green circles represent gut bacteria, and pink circles represent metadata.

**Figure 6 microorganisms-12-00068-f006:**
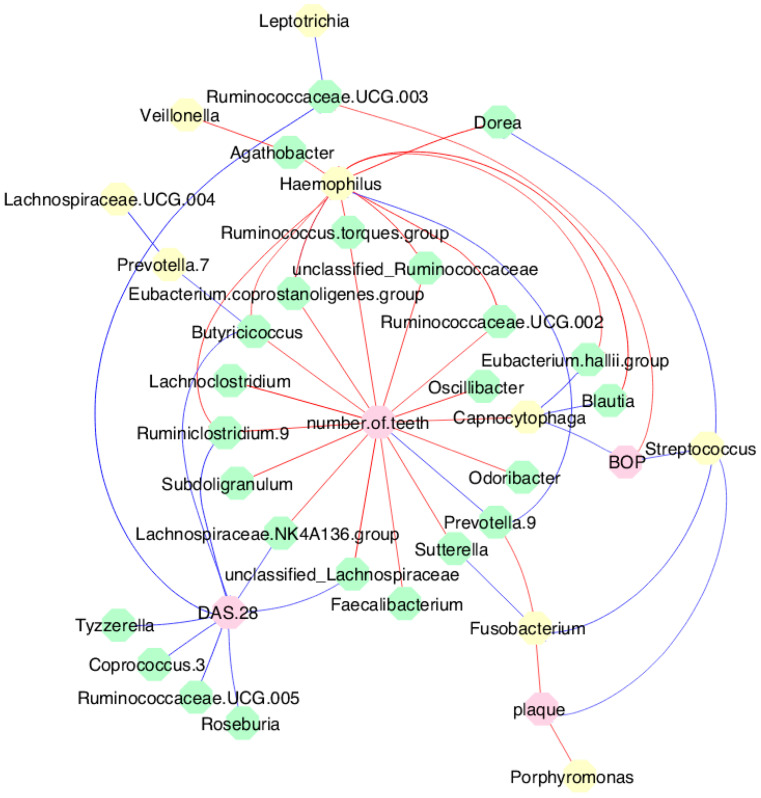
Correlations between oral and gut microbiota and clinical data after MTX treatment. Spearman’s correlation coefficient was used to measure the correlation between bacterial content and quantitative variables. The blue lines represent direct correlations, and the red lines represent inverse correlations. Yellow circles represent oral bacteria, green circles represent gut bacteria, and pink circles represent metadata.

**Table 1 microorganisms-12-00068-t001:** Clinicodemographic data of the individuals with rheumatoid arthritis (RA) included in the study (*n* = 37).

Variables	*n* (%)
**Age, median (min–max)**	51 (22–70)
**Sex**	
Female	33 (89.2)
Male	4 (10.8)
**RA**	
Early	22 (59.5)
Established	15 (40.5)
**Tobacco smoking**	
Never	22 (59.5)
Still	8 (21.6)
Stopped	7 (18.9)
**Alcohol consumption**	
Never	31 (83.8)
Still	2 (5.4)
Stopped	4 (10.8)
**RF (IU/mL), median (min–max)**	15.00 (0–1890)
**ACPA (IU/mL), median (min–max)**	69.75 (0–340)
**CRP (mg/dL), median (min–max)**	0.77 (0–14)
**VAS (mm), median (min–max)**	60 (0–100)
**ESR (mm/h), median (min–max)**	18.00 (2–85)
**DAS 28, median (min–max)**	4.99 (0.63–8.06)
**Disease activity**	
Remission	7 (18.9)
Low	6 (16.2)
Moderate	11 (29.7)
High	13 (35.1)
**MTX treatment * (n, %)**	
Failure	16 (43.2)
Responsive	17 (45.9)
Adverse events	2 (5.4)
NA	2 (5.4)
**Time of treatment ^α^, median (min–max)**	3 (0–408)
**Periodontitis ^∞^ (n, %)**	
None	10 (27.0)
Mild	-
Moderate	21 (56.8)
Severe	6 (16.2)

Note: ACPA: anti-citrullinated protein antibody; CRP: C-reactive protein; DAS 28: disease activity score 28; ESR: erythrocyte sedimentation rate; MTX: methotrexate; NA: not available; RA: Rheumatoid arthritis; RF: rheumatoid factor; VAS: visual analog scale for pain. * Response to MTX treatment assessed at T1. ^α^ Time of treatment in months. ^∞^ Based on the study by Eke et al. [29].

**Table 2 microorganisms-12-00068-t002:** Periodontal parameters, disease activity, and response to methotrexate treatment at T0, T1, and T2.

Variables	T0, *n* = 37	T1, *n* = 32	T2, *n* = 28	*p*-Value
**Probing depth (mm), median (** **min–max** **)**	1.92 (0.86–3.16)	1.70 (0.64–3.70)	1.55 ^a,b^ (0.52–2.98)	<0.001 ^§^
**CAL (mm), median (** **min–max** **)**	2.05 (1.06–5.15)	1.89 (0.83–7.14)	1.83 ^a^ (0.78–6.92)	0.026 ^§^
**BOP (%), median (** **min–max** **)**	9.00 (0–53)	24.00 ^a^ (6–85)	10.00 ^b^ (0–35)	<0.001 ^§^
**Plaque index (%), median (** **min–max** **)**	42.00 (0–100)	37.50 (6–92)	18.00 ^a,b^ (3–50)	<0.001 ^§^
**Periodontitis (n, %)**				
None	10 (27.0)	8 (21.6)	11 (29.7)	<0.05 ^¶^
Mild	0 (0.0)	1 (2.7)	0 (0.0)	
Moderate	21 (56.8)	17 (45.9)	13 ^a,b^ (35.1)	
Severe	6 (16.2)	6 (16.2)	4 (10.8)	
NA	-	5 (13.6)	9 (24.4)	
**DAS 28, median (min** **–** **max)**	4.99 (0.63–8.06)	4.49 (0.14–7.76)	-	0.082 ^§^

Note: BOP: bleeding on probing; CAL: clinical attachment loss; DAS 28: disease activity score 28; NA: not available; T0: first assessment; T1: second assessment; T2: third assessment. ^a^ Difference compared to T0; ^b^ Difference compared to T1. ^§^ Wilcoxon test; ^¶^ Chi-square test.

## Data Availability

Data are contained within the article or Appendix A.

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
