# Peer review of "Methotrexate and Non-Surgical Periodontal Treatment Change the Oral–Gut Microbiota in Rheumatoid Arthritis: A Prospective Cohort Study"

_microorganisms, 2023, doi:10.3390/microorganisms12010068_

Round 1

Reviewer 1 Report (Previous Reviewer 1)

Comments and Suggestions for Authors

A revised version of the manuscript titled 'Methotrexate and non-surgical periodontal treatment change the oral-gut microbiota in rheumatoid arthritis: a longitudinal study' has been submitted to the journal Microorganisms.

While the authors have significantly improved certain aspects of the study, others still require clarification. These are detailed below:

-The title does not accurately convey the type or design of the study; it merely indicates that it is a longitudinal follow-up. Various observational and all experimental studies are characterized by this type of follow-up. Please revise.

-It is necessary to maintain coherence in the terms related to the study's objective as described in the title, the actual objective, and the hypothesis. The title suggests 'change,' the abstract's objective describes 'influence,' the introduction's objective mentions 'effect,' and the hypothesis describes 'impact.' Please review and adjust for coherence.

-The abstract does not include some relevant results described in the main text that present significant values, including certain correlations. In the case of correlations, please provide the magnitude of the correlation (weak, moderate, strong), its direction, along with the p-value.

-In the abstract, modified interactions between clinical parameters and the microbiota are also described; however, the results do not show findings related to interaction. The statistical analysis does not describe the tests used to evaluate interaction, and in the results of the main text, no interaction findings are reported. Please revise.

-In the methodology, it is necessary to clarify the type of study based on the recommendations mentioned above.

-In Figure 1, it is recommended to include the number of patients with and without periodontitis throughout the entire follow-up.

-Also, in the methodology, the reasons for using two classifications for periodontitis diagnoses are not clear. This could potentially create confusion for readers. If the authors believe it is essential to maintain both classifications, the discussion should provide the reasons for doing so.

-The Spearman test described in the data analysis should be shifted to the statistical analysis.

-The chi-square test should not be considered a non-parametric test, as indicated in parentheses.

-In sections 3.6 and 3.7, some statistically significant correlations are presented. Please provide the magnitude of the correlation, its direction, and the p-value. Based on these results, it is possible that the authors may need to adjust the discussion, as some correlations may be weak or moderate. Please review.

-In the captions of Figures 5 and 6, intercorrelations are described. What is the meaning of this term? Statistically speaking, this term is not common. Please revise.

-An additional limitation is related to the study design.

Comments on the Quality of English Language

minor

Author Response

Reviewer: #1

While the authors have significantly improved certain aspects of the study, others still require clarification. These are detailed below.

Response: We would like to thank the reviewer for the attention and the time taken to provide constructive comments to improve the quality of our manuscript. Detailed responses to the concerns raised are given below.

-The title does not accurately convey the type or design of the study; it merely indicates that it is a longitudinal follow-up. Various observational and all experimental studies are characterized by this type of follow-up. Please revise.

Response: The purpose of the present study was to investigate the changes on the oral-gut microbiota composition of individuals with rheumatoid arthritis caused by methotrexate and non-surgical periodontal treatment. In this study, individuals were exposed to methotrexate and non-surgical periodontal treatment and followed-up. Therefore, a defined group of people (the cohort) was followed over time, to examine associations between different interventions received and subsequent outcomes, as previously reported elsewhere [1, 2, 3]. According Mathes et al. [4] “all studies with exposure-based sampling gather multiple exposures (with at least two different exposures or levels of exposure) should be considered cohort studies”. The title was modified accordingly.

“Methotrexate and non-surgical periodontal treatment change the oral-gut microbiota in rheumatoid arthritis: a prospective cohort study”

References:

  1. Millen AE, Dahhan R, Freudenheim JL, Hovey KM, Li L, McSkimming DI, Andrews CA, Buck MJ, LaMonte MJ, Kirkwood KL, Sun Y, Murugaiyan V, Tsompana M, Wactawski-Wende J. Dietary carbohydrate intake is associated with the subgingival plaque oral microbiome abundance and diversity in a cohort of postmenopausal women. Sci Rep. 2022 Feb 16;12(1):2643. doi: 10.1038/s41598-022-06421-2.

2 Dashper SG, Mitchell HL, Lê Cao KA, Carpenter L, Gussy MG, Calache H, Gladman SL, Bulach DM, Hoffmann B, Catmull DV, Pruilh S, Johnson S, Gibbs L, Amezdroz E, Bhatnagar U, Seemann T, Mnatzaganian G, Manton DJ, Reynolds EC. Temporal development of the oral microbiome and prediction of early childhood caries. Sci Rep. 2019 Dec 24;9(1):19732. doi: 10.1038/s41598-019-56233-0.

3 Uchida-Fukuhara Y, Ekuni D, Islam MM, Kataoka K, Taniguchi-Tabata A, Fukuhara D, Toyama N, Kobayashi T, Fujimori K, Sawada N, Iwasaki Y, Morita M. Caries Increment and Salivary Microbiome during University Life: A Prospective Cohort Study. Int J Environ Res Public Health. 2020 May 25;17(10):3713. doi: 10.3390/ijerph17103713.

4 Mathes T, Pieper D. Clarifying the distinction between case series and cohort studies in systematic reviews of comparative studies: potential impact on body of evidence and workload. BMC Med Res Methodol. 2017;17(1):107.

-It is necessary to maintain coherence in the terms related to the study's objective as described in the title, the actual objective, and the hypothesis. The title suggests 'change,' the abstract's objective describes 'influence,' the introduction's objective mentions 'effect,' and the hypothesis describes 'impact.' Please review and adjust for coherence.

Response: As suggested by the reviewer, the objective described in paragraph 1 of the “Abstract” section (page 3) was amended as follows:

“This study evaluated the changes in the composition of oral-gut microbiota in patients with rheumatoid arthritis (RA) caused by methotrexate (MTX) and non-surgical periodontal treatment (NSPT).”

Paragraph 5 of the “Introduction” section (page 5) was also amended, as follows:

“The purpose of the present longitudinal study was to investigate the changes on the oral-gut microbiota composition of individuals with RA caused by MTX and NSPT and their relationship with clinical and biochemical parameters. Our hypothesis was that MTX and NSPT would change the diversity of the oral-gut microbiota.”

-The abstract does not include some relevant results described in the main text that present significant values, including certain correlations. In the case of correlations, please provide the magnitude of the correlation (weak, moderate, strong), its direction, along with the p-value.

Response: The “Abstract” section should be restricted to 200 words, which considerably limits the detailing of information. However, to address the reviewer's concerns, information about the results of the correlations was added, as follows:

“The results of the analysis of continuous variables were presented descriptively and non-parametric tests and Spearman's correlation were adopted. Thirty-seven patients (27 with periodontitis) were evaluated at T0; 32 patients (24 with periodontitis) at T1; and 28 patients (17 with periodontitis) at T2. MTX tended to reduce the alpha diversity of the oral-gut microbiota, while NSPT appeared to increase the number of different species of oral microbiota. MTX and NSPT influenced beta diversity in the oral microbiota. The relative abundance of oral microbiota was directly influenced by periodontal status. MTX did not affect the periodontal condition but modified the correlations that varied from weak to moderate (pË‚0.05) between clinical parameters and the microbiota. MTX and NSPT directly affected the composition and richness of the oral-gut microbiota. However, MTX did not influence periodontal parameters.”

-In the abstract, modified interactions between clinical parameters and the microbiota are also described; however, the results do not show findings related to interaction. The statistical analysis does not describe the tests used to evaluate interaction, and in the results of the main text, no interaction findings are reported. Please revise.

Response: The authors apologize for this oversight. The results cited in the abstract as “interactions” correspond to the correlations described in the “3.6. Periodontal parameters and RA activity influenced the oral and gut microbiota” and “3.7. MTX treatment defined new associations of periodontal and RA parameters with oral and gut bacteria” subsection of the “Results” section (page 13-16), calculated using Spearman's correlation coefficient. As suggested by the reviewer, the correlation magnitude, direction, and p-value were added in these subsections and the term “interactions” was corrected in the “Abstract” section (page 3), as follows:

“MTX did not affect the periodontal condition but modified the correlations that varied from weak to moderate (pË‚0.05) between clinical parameters and the microbiota.”

-In the methodology, it is necessary to clarify the type of study based on the recommendations mentioned above.

Response: We thank the reviewer`s comments. As suggested and according to what had been discussed above, the design of the study was defined and added in paragraph 1 of “2.1. Study design, setting, and ethical clearance” subsection of the “2. Materials and Methods” section (page 6), as follows:

“This prospective cohort study performed at the Rheumatology Outpatient Clinic of the Ribeirão Preto Medical School Hospital, São Paulo, Brazil, between August 2018 and December 2019.”

-In Figure 1, it is recommended to include the number of patients with and without periodontitis throughout the entire follow-up.

Response: As suggested, Figure 1 was altered and the number of individuals with and without periodontitis was added at the three follow-up times.

-Also, in the methodology, the reasons for using two classifications for periodontitis diagnoses are not clear. This could potentially create confusion for readers. If the authors believe it is essential to maintain both classifications, the discussion should provide the reasons for doing so.

Response: The classification of periodontitis according to the criteria of Tonetti et al. (2018) was removed to make the manuscript clear and in a manner that it does not impact the observed results. Therefore, only the classification of periodontitis according to the criteria of Eke et al. (2012) remained, allowing for an analysis according to the severity of periodontitis.

-The Spearman test described in the data analysis should be shifted to the statistical analysis.

Response: As suggested, the Spearman's test was added in the “2.7. Statistical analysis” subsection of the “Materials and Methods” section (page 10-11).

-The chi-square test should not be considered a non-parametric test, as indicated in parentheses.

Response: The reviewer is right. The sentence has been amended in the “2.7. Statistical analysis” subsection of the “Materials and Methods” section (page 10-11), as follows:

“The Statistical Package for the Social Sciences (SPSS) (version 25.0, Armonk, USA) and the GraphPad Prism (version 7.00, La Jolla, CA, USA) were employed for the statistical analyses. Descriptive analysis was applied to clinicodemographic aspects and periodontal data. The Shapiro-Wilk test revealed that continuous data exhibited a non-normal distribution (p<0.001). Bivariate analysis involving continuous data was performed with the Wilcoxon test. The results of the analysis were presented as median, minimum, and maximum. Chi-square test was adopted for bivariate analysis involving categorical data. Spearman's correlation coefficient was used to measure the correlation between bacterial content and quantitative variables. For all analyses, the level of significance was set at <0.05.”

-In sections 3.6 and 3.7, some statistically significant correlations are presented. Please provide the magnitude of the correlation, its direction, and the p-value. Based on these results, it is possible that the authors may need to adjust the discussion, as some correlations may be weak or moderate. Please review.

Response: As suggested by the reviewer, the correlation magnitude, direction, and p-value were added in subsection “3.6. Periodontal parameters and RA activity influenced the oral and gut microbiota” and “3.7. MTX treatment defined new associations of periodontal and RA parameters with oral and gut bacteria” of the “Results” section (pages 13-16). Additionally, adjustments to the "Discussion" section (pages 17-18) were made.

-In the captions of Figures 5 and 6, intercorrelations are described. What is the meaning of this term? Statistically speaking, this term is not common. Please revise.

Response: Figures 5 and 6 demonstrate the correlations between oral and gut bacteria and periodontal and rheumatological clinical parameters before and after treatment with methotrexate, calculated using Spearman's correlation coefficient. The term “intercorrelations” has been replaced by “correlations” in the legends of Figures 5 and 6 (pages 32-33).

-An additional limitation is related to the study design.

Response:. A statement on this issue has been included in the last paragraph of the “Discussion” section (page 20), as follows:

The current study has shortcomings that should be acknowledged. The first concerns the absence of a control group of individuals who had not been exposed to MTX and NSPT for comparison, precluding the conclusion that changes in oral-gut microbiota were due solely to the exposures being studied, rather than to other factors.”

Reviewer 2 Report (Previous Reviewer 2)

Comments and Suggestions for Authors

The manuscript has been improved

Author Response

Reviewer: #2

The manuscript has been improved

Response: We would like to thank the reviewer for his/her attention.

Round 2

Reviewer 1 Report (Previous Reviewer 1)

Comments and Suggestions for Authors

The authors have made the corrections appropriately, therefore publication of this manuscript is recommended.

Comments on the Quality of English Language

Minor

This manuscript is a resubmission of an earlier submission. The following is a list of the peer review reports and author responses from that submission.

Round 1

Reviewer 1 Report

Comments and Suggestions for Authors

The manuscript “Methotrexate regimen and periodontal therapy change the oral gut microbiota composition in rheumatoid arthritis” was submitted to Microorganisms.

This study evaluated the influence of methotrexate and periodontal therapy on the composition of the oral gut microbiota of individuals with rheumatoid arthritis. The authors concluded that methotrexate and periodontal treatment directly affected the composition and richness of the oral gut microbiota, with the emergence of different bacterial and microbial group interactions.

The authors deal with an interesting topic; however, some adjustments must be considered.

Title: Please include the type of study.

Abstract

The number of patients diagnosed with periodontitis must be presented. Loss to follow-up must be recognized. The conclusions are based on major limitations of the study. Negative results must also be recognized, and conclusions drawn about them.

Keywords: Please make sure they are MeSH terms.

This reviewer is not sure about the need to include a section called "importance" in this journal. If not, this comment should be removed.

Introduction

In paragraphs 1 and 2 the authors are very emphatic with favorable results; however, objectivity is recommended since there are contradictory results. Moreover, it is recommended to cite the best available evidence. Some statements also present too many references.

Periodontitis and its holistic commitment to the problem that the authors wish to elucidate must be defined. The process is more complex than the participation of some bacteria and their deleterious effects.

Regarding the problem raised, not only one study (reference 10) shows results in this regard. Other studies have been published on this matter; therefore, the authors must present a state-of-the-art related to the results that exist so far. Subsequently, the gap in knowledge that this study aims to find must be indicated.

Methods

For clarity, the selection criteria should be mentioned in the main text. Authors should remember that the journal has no word limit in its main text.

The results of the examiners' calibration and the statistical test used must be presented.

Please clarify the diagnostic references of periodontitis. References 40 and 41 include the current classification and the immediately previous one. Besides, as previously recommended, details indicated to be included in a supplementary file should be presented in the main text. Consider this recommendation in the rest of the methodology.

Check the spelling of "planning".

Was the normal distribution of the data assessed? If so, present the test and its p-value. It was not indicated that correlations were made (Spearman). Besides, the use of non-parametric tests is not understood when it is also indicated that ANOVA was used.

A sample size calculation is essential. Only 24 patients with periodontitis were evaluated and 17 were re-evaluated.

The absence of different control groups is a shortcoming that must be presented.

Results

MAJOR CONCERN: The reasons for including patients diagnosed with different classifications and the implications of this should be explained in detail. Furthermore, the reasons for performing periodontal treatment in periodontally healthy patients (37%) should be explained. Is this ethical?

Table 1. Please revise the percentage of healthy patients. The data is wrong.

Table 2. p≤0.05. Please revise ≤.  What does NA mean in Table 2? Are patients lost to follow-up? This should be indicated in the text.

Finally, it appears that only 24 patients with periodontitis were treated (T1), and only 17 patients initially evaluated with periodontitis were re-evaluated (T2). This should be described with emphasis in the text (MAJOR CONCERN). Reasons for loss to follow-up must be presented.

The results of the described correlations must be presented with their respective p-values.

Figure 5 is very confusing.

Discussion

In the present study, there was no effect of MTX on the periodontium. The small sample size has a great influence on the results of this study. This and other results with the same trend are due to the same reason.

This study must recognize the great number of limitations that it presents. It is necessary to clarify many shortcomings that were described above.

Conclusions

First, it must be recognized that this study concludes based on major limitations. Negative results should also be presented in the conclusions.

Comments on the Quality of English Language

moderate editing

Reviewer 2 Report

Comments and Suggestions for Authors

The content of the submitted manuscript is good but the presentation way of current form is not fulfilling the journal requirements. Modification is needed to consider for publication.

Please see enclosed pdf for a point by point analysis

Comments on the Quality of English Language

English is fine